# Effect of Leukoreduced Platelet Rich Plasma on Intra-Articular Pro-Inflammatory Cytokines in a Canine Pilot Study

**DOI:** 10.3390/ani12172163

**Published:** 2022-08-24

**Authors:** J. Alberto Gines

**Affiliations:** Department of Veterinary Clinical Sciences, School of Veterinary Medicine, Louisiana State University, Baton Rouge, LA 70803, USA; agines1@lsu.edu

**Keywords:** platelet rich plasma, cytokines, dog, stifle, osteoarthritis

## Abstract

**Simple Summary:**

This study provides initial information on the effect of platelet rich plasma in some of the most important pro-inflammatory cytokines identified in osteoarthritic stifle joint in dogs. The use of platelet rich plasma is getting more popular in the veterinary field but its effect in those cytokines is not known.

**Abstract:**

Evaluation of effect of Leukoreduced Platelet Rich Plasma (LrPRP) on TNF-α and IL-6 (pro-inflammatory cytokines) in joint fluid in dogs with cranial cruciate ligament rupture (CCLR). Eight client-owned dogs with CCLR were assigned to treatment (2 mL LrPRP) or control (2 mL saline) injection groups. Day of evaluation (day 0) and day of surgery (day 10–14), joint fluid was collected and joint injected. Joint fluid was also collected on day of suture removal (day 20–28). TNF-α and IL-6 concentrations of joint fluid were measured using a bead-based antibody assay. Concentrations at the later time points were expressed as a ratio to the initial level within each stifle. LrPRP had a mean concentration of platelets 1.7 times higher that of whole blood on day of evaluation and 1.4 times higher on day of the surgery. Leukocytes were reduced by 99.7%. On day of surgery, TNF-α ratios in the joint fluid from dogs injected with LrPRP were significantly different that TNF-α ratios of control group. On the day of suture removal ratios of IL-6 and TNF-α were lower in LrPRP treatment group compared with control group; however, differences were not significant. LrPRP modulate ratios of pro-inflammatory cytokine TNF-α in dogs with CCLR.

## 1. Introduction

The use of platelet-rich plasma (PRP) to treat a variety of inflammatory and degenerative disorders, including those of the musculoskeletal system [1,2,3,4,5,6,7,8,9,10,11,12,13,14,15,16], is increasing. One of the most common uses of PRP is the intra-articular administration in cases of osteoarthritis (OA) [2,3,4,5,15,16,17,18,19]. The principle behind this treatment is to concentrate platelets from fresh whole blood while removing most of the other cell types. When injected, an elevated concentration of growth factors present in the platelet α-granules are released, with the goal of reducing inflammation and enhancing healing.

It is unknown which concentration of platelets in the PRP are the most appropriate but the cellular composition of the PRP appears to influence its biologic effect. Platelets increase the anabolic signaling and the leukocytes the catabolic environment and both can influence the biological effect of the PRP [20].

OA is associated with cartilage damage, bone remodeling and synovitis of the joint. The development and progression of OA is associated with the presence of inflammation [21]. The main pro-inflammatory cytokines involved in the pathophysiology of the OA have been defined as TNF-α, IL-1β and IL-6 [22,23]. There is evidence that leukocyte-reduced PRP (LrPRP) has some efficacy in the treatment of OA by decreasing pain and inflammation [3,17,19] when compared with untreated patients. However, it is unclear if LrPRP has any effect on the pro-inflammatory cytokines present in the joint fluid of joints with OA. It is known that the concentration of these pro-inflammatory cytokines is elevated in dogs with natural occurring cranial cruciate ligament (CCL) rupture when compared with normal stifles [24,25].

The objective of this study is to evaluate the concentration of pro-inflammatory cytokines in dogs with stifle OA and how intra-articular administration of LrPRP affects that concentration. We hypothesized that administration of intra-articular LrPRP will have an effect on the pro-inflammatory cytokines TNF-α and IL-6, present in the stifle joint of dogs with CCL rupture, when compared with administration of saline.

## 2. Materials and Methods

This study was designed as a randomized, controlled trial and was approved by the Institutional Animal Care and Use Committee at North Carolina State University (IACUC ID 16-044-O). All clients signed a consent form prior to enrolment.

### 2.1. Inclusion Criteria

Client-owned dogs with history of hind limb lameness and diagnosis of CCL rupture were included when a tibial osteotomy procedure was elected by the owner as a method of treatment. Complete history and full orthopedic exam were performed at the time of evaluation. 

Age, sex, breed, lameness duration and prior use of NSAIDs were recorded.

### 2.2. Study Protocol

A simple size calculation was performed using a power of 0.8 with a 0.05 alpha and a difference between groups of 40%, which showed a minimum of 4 dogs per group. A random table for 8 dogs was created using an online random number generator (www.random.org, accessed on 6 July 2022). Dogs were assigned to the treatment or control group based on this table. Each dog was seen the day of evaluation (Day 0), day of the surgery (Surgery—day 10–14) and the day of recheck or suture removal (Recheck—day 20–28).

On the day of evaluation (Day 0), dogs were sedated with dexmedetomidine (3 mcg/kg IV) and hydromorphone (0.1 mg/kg IV) and radiographs were performed for surgical planning. The affected joint was prepared for sterile joint fluid aspiration and injection by clipping the hair and sterile preparation of the skin. After joint fluid was aspirated, 2 mL of either LrPRP or saline was injected, depending on the assigned group. On the day of the surgery (Surgery), joint aspiration was performed after the limb was prepared and draped. A medial mini-arthrotomy was performed to inspect the joint, and remove visible remnants of the CCL, and damaged portions of the medial meniscus. The status of the CCL (partial or complete tear) and the medial meniscus (presence or not of meniscal tear) was recorded. A tibial osteotomy procedure was performed (tibial plateau leveling osteotomy -TPLO-, triple tibial osteotomy -TTO-, or cranial closing wedge osteotomy -CCWO-). The election of the procedure was based on surgeon preference or previous surgery performed in the contralateral stifle. Once the procedure was completed and the joint capsule closed, the joint was injected with 2 mL of LrPRP or saline, depending on their treatment group assignment. All dogs were discharged with a 5 day course of cephalexin and a short course of carprofen.

On the day of the suture removal (Recheck) dogs were sedated with dexmedetomidine and hydromorphone, the joint area prepared, and joint fluid was aspirated from the lateral side of the patellar ligament. All stifles were injected with 2 mL of LrPRP.

### 2.3. Joint Fluid Sample and Cytokine Analysis

Once the joint fluid was obtained it was immediately placed on ice. The sample was placed into one or two aliquots (depend on initial volume) and, within 30 min of collection, centrifuged at 1000× *g* for X min to eliminate the cells present in the sample. The cell-free fluid was separated and stored at −80 °C until further analysis. Cytokine analysis of TNF-α and IL-6 was performed using a Milliplex^MAP^ Canine Cytokine Magnetic Bead Panel kit (EMD Millipore, Burlington, MA, USA).

### 2.4. PRP Preparation and Analysis

LrPRP was prepared using ACP (autologous conditioned plasma, Arthrex, Naples FL) syringes. Jugular venipuncture was performed using a 19 G butterfly needle and 14 mL of whole blood was draw into the ACP syringe that already contained 1.5 mL of the anticoagulant ACD-A (Anticoagulant Citrate Dextrose Solution, Solution A). One mL of the whole blood sample was placed in an EDTA tube for automatic cell count analysis and smear evaluation. The remaining 14.5 mL were centrifuged at 1500 rpm for 5 min. One mL of the LrPRP was placed in an EDTA tube for automatic cell count analysis and smear evaluation. Only the LrPRP samples for day of evaluation and day of surgery were sent for cell count analysis and smear evaluation.

The blood sample from the dogs receiving LrPRP at the time of surgery was collected at the end of the surgical procedure before closure of the surgical incision and the LrPRP immediately prepared and administered.

### 2.5. Statistical Analysis

The levels of TNF-α and IL-6 at day of surgery and at day of suture removal were expressed as ratios of the level obtained in relation of the level on day of evaluation to normalize the data for each dog. The ratios were compared over treatment within time using U Mann–Whitney test (JMP Pro 13.0.0, SAS Institute Inc., Cary, NC, USA). A *p* < 0.05 was considered statistically significant.

## 3. Results

The age of the dogs ranged from 2 to 13 years with a median of 6 years, 5 were female spayed and 3 male castrated. The breed of the dogs was 2 Labrador retriever, and 1 Pitbull mix, Rottweiler, American Staffordshire Terrier, Boxer mix, English Spaniel and Mastiff mix. The duration of the lameness ranged from 1.5 to 12 months with a median of 4 months. None of the dogs were giving NSAIDs two weeks prior presentation on day 0. 

The surgical procedure performed was TPLO in 5 cases, TTO in 2 and CCWO in 1 case. Meniscal tear was present in 3 dogs on each group and partial meniscectomy was performed on those cases. All dogs were discharged with a short course (median 8.5 days) of carprofen (2.2 mg/kg orally, twice daily).

### 3.1. PRP Analysis

Results of the analysis of the whole blood and LrPRP at day of evaluation and day of surgery are summarized on Table 1. 

The volume of LrPRP obtained at day of evaluation was 3.5 mL (range 3 mL to 4.5 mL), and the volume at day of surgery was 4.5 mL (range 3.5 mL to >6 mL). The platelet concentration obtained in the LrPRP at day of evaluation was 1.7 times higher that of whole blood and at day of surgery 1.4 times higher compared with whole blood. The reduction of leukocytes was confirmed in all samples, with a platelet/WBC ratio of 34 on whole blood and 3940 on LrPRP the day of evaluation and 47.5 on whole blood and 2507 on day of surgery. A 99.7% leukoreduction was achieved on both days (day of evaluation and day of surgery) when whole blood and LrPRP were compared.

### 3.2. Cytokine Analysis

The ratio of the cytokine levels at day of surgery and day of suture removal to those obtained on day of evaluation are displayed in Figure 1 and Figure 2. On the day of surgery, the TNF-α ratios were significantly different between control and LrPRP group (*p* = 0.0421) and the TNF-α ratios were lower in the LrPRP group.

The TNF-α ratio for the LrPRP group on the day of suture removal was not statistically different from the TNF-α ratio of control group (*p* = 0.1124).

No significant differences were present for the IL-6 ratios, between LrPRP and control groups.

The mean values for the different ratios were always lower for the LrPRP group than the control group for each measurement day (day of surgery and suture removal) for TNF-α and IL-6 (Figure 1 and Figure 2).

## 4. Discussion

This study has demonstrated a significant difference on TNF-α ratios between LrPRP and control groups on the day of surgery (10 days after LrPRP injection) in dogs with natural occurring CCL rupture. Additionally, the TNF-α ratios at the time of suture removal were close to be statistically significant. There was no difference in the ratio of IL-6 over time between the LrPRP and control groups. 

The platelet concentration present in PRP has been always the main factor in the definition of PRP, however the optimum platelet concentration to provide a benefit for intra-articular injection is unknown. Using PRP with a concentration of 3 times the number present in whole blood appears to be effective [26], but lower concentrations have been showed to have similar results [3,5]. The LrPRP in this study had a concentration of around 1.5, which may have impacted its ability to reduce the inflammatory response. The concentration of platelets was inferior on the day of surgery due to hemodilution as the blood sample was obtained at the end of the surgical procedure and to obtain of the platelet rich plasma the manufacturer instructions were followed. Discharging the platelet poor plasma before obtaining the platelet rich plasma should provide a higher platelet concentration.

Different systems will provide PRP with a different cellular composition [27,28,29]. While leukocyte-rich PRP have showed to have more pro-inflammatory properties [30], LrPRP appears to stimulate matrix anabolism [20,30]. The leukocyte concentration of the PRP affects his efficacy on the treatment of OA, with LrPRP improving functional outcomes in patients with knee OA when compared with leukocyte rich PRP, hyaluronic acid or saline [12]. The leukoreduction achieved on the PRP in this study was high with a 99.7% reduction at both collection times.

How often intra-articular injection of PRP must be repeated or how long a single injection is effective, are questions still not resolved [31]. In the design of this study, injection was performed with intervals of 10–14 days. It is described that when collagen is used as a platelet activator, the release of the anabolic cytokines can be present for 7 days [32]. In humans, the use of LrPRP at the time of knee surgery has showed better recovery and less pain than the control group [17]. In the present study, the use of LrPRP at the time of the surgical procedure demonstrate a lower mean ratio for both cytokines evaluated when they are compared with the control group on the day of suture removal.

It has been demonstrated that the nonsteroidal anti-inflammatory drugs (NSAID) (including carprofen) do not affect the platelet aggregation and the release of the growth factors so the discontinuation of this NSAID before PRP production and use is not considered necessary [33]. In the current study, carprofen was prescribed after the surgical procedure to all dogs with a mean duration time of 8.5 days. The ratio for both cytokines was lower for the LrPRP compared with the control group on the day of suture removal, which indicate that carprofen did not affect the LrPRP in the present study.

Of the two pro-inflammatory cytokines evaluated, LrPRP had a higher effect in this study over the TNF-α by decreasing the ratios in the LrPRP group compared with the control group. There is evidence that blocking the production of TNF-α could counteract the degradative mechanisms related with OA [22]. Further studies are required to evaluate the relationship between modulation of the TNF-α levels with clinical improvement in dogs, however gait improvement of dogs with CCLR have been reported when single intra-articular stifle injection was given in non-surgical patients [16].

There are some limitations on this study. The most important is that this is the low number of cases, and this could be the reason that no significant differences were found between LrPRP and saline groups for the IL-6. It is unknown the duration of the anabolic effect of the growth factors and cytokines present in the platelets, and how this effect is influenced by the platelet concentration. The different of LrPRP composition seen at day of presentation and day of surgery could potentially affect the effect seen on cytokines values. Another limitation is the degree of inflammation on those stifles and the effect of the meniscal tears over the values of the pro-inflammatory cytokines. Different surgeons were performing the surgical procedures and this could influence the inflammation of the tissues, however all surgeons were specialists and they have a long surgical experience. 

## 5. Conclusions

In conclusion, the use of LrPRP platelet rich plasma has a positive impact on TNF-α by reducing the ratios of this pro-inflammatory cytokine between LrPRP or saline intra-articular injections in dogs with natural occurring CCL rupture. Additionally, there was a non-significant reduction in the IL-6 ratios in the LrPRP treated dogs when compared with control group. Further studies are required to correlate these findings with functional use of the affected limb and effect on the length of recovery.

## Figures and Tables

**Figure 1 animals-12-02163-f001:**
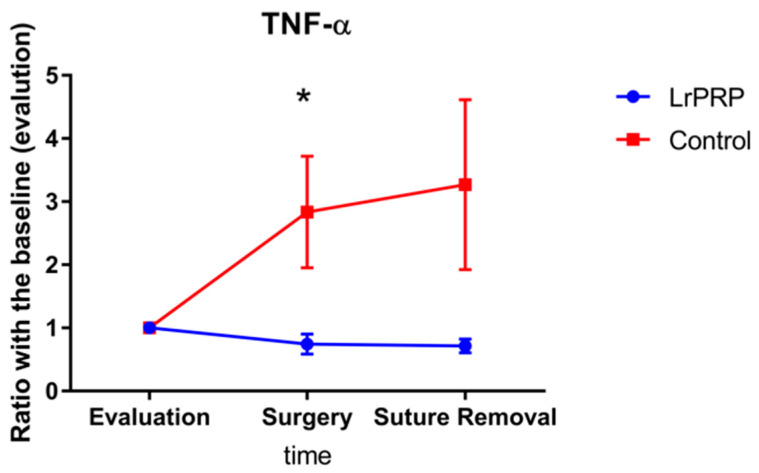
Evolution of TNF-a between intra-articular injection of LrPRP group and the saline group at the time of day of surgery and day of suture removal when compared with day of evaluation. * Significant difference between groups *p* = 0.0294.

**Figure 2 animals-12-02163-f002:**
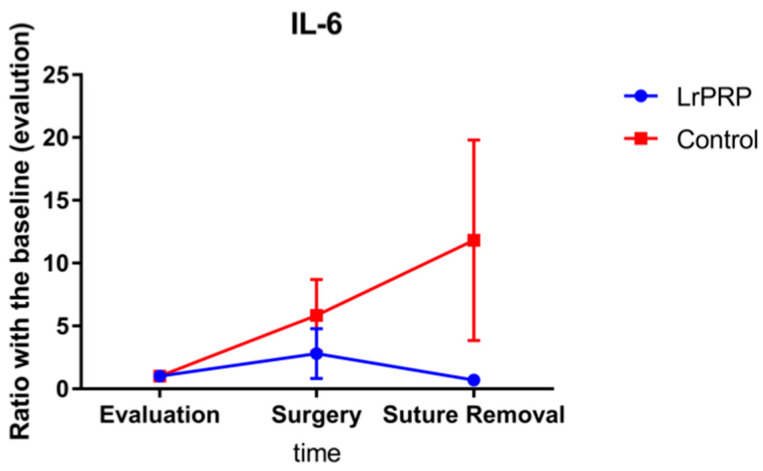
Evolution of IL-6 between intra-articular injection of LrPRP group and the saline group at the time of day of surgery and day of suture removal when compared with day of evaluation.

**Table 1 animals-12-02163-t001:** Results of the cell count analysis of blood samples and LrPRP on the day of evaluation and day of surgery.

Blood Cells	Day of Evaluation		Day of Surgery	
	Whole Blood	LrPRP	Whole Blood	LrPRP
Platelets (×10^3^/μL)	255	433.5	254.5	362
WBC (×10^3^/μL)	8.09	0.11	5.68	0.12
Neutrophils (×10^3^/μL)	4.98	0.02	3.57	0.03
RBC (×10^3^/μL)	5.74	0.04	5.06	0.04
Platelets/WBC ratio	34	3940	47.5	2507

## Data Availability

The original contributions presented in the study are included in the article, further inquiries can be directed to the corresponding author.

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
