# Peer review of "Effect of Leukoreduced Platelet Rich Plasma on Intra-Articular Pro-Inflammatory Cytokines in a Canine Pilot Study"

_animals, 2022, doi:10.3390/ani12172163_

Round 1

Reviewer 1 Report

Study compares 4 (LrPRP) vs. 4 (Saline) CCLR intraarticular joint fluid analysis for pro-inflammatory cytokines from pre-surgery, at surgery, and at suture removal. The findings are encouraging, but an increased case number are needed to establish if the cytokine changes have any clinical outcomes (pain scores, lameness, time to recovery, etc.). Overall the article is well written, presented clearly, and findings are stated appropriately. 

Comments:

Suggested Title: Effect of Leukoreduced Platelet Rich Plasma on Intra-Articular Pro-Inflammatory Cytokines in a Canine Pilot

Need to include Lr in the title as PRP alone may not yield the same results.

This study is a pilot  as there are too many variables to stipulate that the treatment with LrPRP alone is the source of the outcome given an n=4. Eight client owned subjects (4 LrPRP vs 4 saline). Of the split groups, it is unclear which received the specific surgery (TPO vs. TTO vs. CCWO) and which had the 3 meniscal tears – These details should be included somewhere within the article as it correlates to the specific treatment group. Duration of lameness ranged from 1.5 to 12 months (median 4 months) – Again, a break down by treatment group in a table or otherwise would be helpful to see if there were any differences. The same is true for prior type of NSAID used and duration. All of which can alter the starting physiology of the joint.  

Line 54: What about blinding?

Line 68: Single or multiple orthopedic surgeons? – If multiple, need to mention as a possible limitation due to tissue handling differences between operators.  

Line 115: What about the duration and types of prior NSAID use as mentioned in line 62?

Line 124: What was the Platelet, WBC, Neutrophil, RBC and Platelet/WBC ratio for the 4 dogs that received Saline control? Was the systemic blood sample composition comparable between study groups?

Line 133: In the intro, you state that “the main pro-inflammatory cytokines involved in the pathophysiology of the OA have been defined as TNF-alpha, IL-1beta, and IL-6,” is there data for IL-1beta or an explanation for not including in the analysis?

Line 173: “his” – the? or delete

Reviewer 2 Report

This is a study of interest on the effect that a PRP low in leukocytes can exert on some intra-articular cytokines in dogs with stifle OA. It shows a simple and appropriate design to determine the changes that the authors look for. However, it has important limitations that affect the validity of the conclusions. The main one is the low number of patients (4 per group) that directly affects the statistical power of the work, and therefore, the validity of the results. This influences the lack of statistical significance in the comparisons (with the exception of TNF in surgery) although the authors' analysis focuses on the trend, something that is not entirely correct. Therefore, my recommendation would be that the authors increase the sample size of this study, in order to assess real conclusions, and resubmit the study. Some specific comments appear below.

Line 62. Was prior NSAIDs use an exclusion criterion, or just depending on the time elapsed between dosing and PRP administration? If a minimum time between actions was considered, this should be provided.

Line 64. How was the sample size of 8 dogs calculated? This information should be provided since it could influence in the results.

Line 71. Was the blood sample to prepare the PRP taken with the patient sedated or awake? Sedation can influence the platelet count obtained in the sample, and therefore the platelet concentrate obtained.

Line 86. If until this moment (recheck) the patients were divided between PRP (n=4) or saline (n=4), why is everyone receiving PRP now?

Line 103-105. Why do you take the sample at the end of the surgery, and not just before the sedation?

Line 107. Why do they make a ratio instead of expressing it in units (pg/mL)?

Line 109. With this number of patients, a non parametric test (U Mann-Whitney) should be used it. Please, correct it because your statistical significance in your test could change.

Line 115-116. With this range of clinical duration of lameness, the degree of inflammation present in the stifles of each animal could be influenced for this issue, and this could influence the levels of cytokines or their response to PRP. Please explain why this situation was not normalized, or why the study was not designed to control for this factor.

130-131. This sentence is redundant information of the table and maybe, it could be removed it.

Line 135-136. The graph shows just the opposite, that the TNF-α ratio is lower in LrPRP. It is always advisable to provide the difference with its 95% confidence interval with the p value.

Line 139. Please, include “different from the TNF-α ratio of control group”. Seeing the variability observed in the values at the time of suture removal, it is likely that a low statistical power would have influenced the lack of statistical significance.

Line 164-169. The fact of counting at different times may clearly be the factor that determined the difference in values, and since the design was established a priori by the researchers, it is therefore important to define this decision, which has partially modified the characteristics of the PRP administered between both treatments.

Line 184-186. Although the graph shows that the values are lower, in reality no significant differences have been detected in either of the two cytokines, so it cannot be concluded that the administration of PRP on the day of surgery has reduced the levels. This analysis may simply be due to the small sample size.

Line 192-194. See comments previously.

Reviewer 3 Report

The original article entitled “Effect of Platelet Rich Plasma on intra-articular pro-inflammatory cytokines in dogs” is very interesting and is well designed and organized. However, authors should address the comments below before considering for publication.

General considerations

·       The references must be improved. No articles published in the last 5 years have been included, so in my opinion the references should be actualized.

Abstract

·       Lines 12 and 13: the abbreviation CCLR is repeated, please, only use it after “rupture”.

·       Lines 13 and 14: the word milliliters should be abbreviated as mL or ml instead of “mls” (according to International Metric System). Please review the text because you have used “mls” several times.

·       Lines 18 and 19: specify that the mean concentration of LrPRP is higher than those of whole blood.

Introduction

·       Line 49: change the word “osteoarthritis” by its abbreviation “OA”, as it has been previously defined.

Materials and methods

·       Inclusion criteria: In the introduction (lines 48-50) you mention than the objective of the study is to evaluate the concentration of proinflammatory cytokines in dogs with stifle OA. However, in the inclusion criteria, you have not mentioned that the included dogs were diagnosed with OA. If all dogs included had OA, I think than you should specify it as well as describe the OA degree of all these dogs, were all dogs severely affected? Or mildly affected? In case that animals with different degrees of OA were included, was that take into consideration when analyzing TNF-α and IL-6 concentration? In this sense, the analysis of the individual results can be interesting in addition to evaluating the averages, as it allows discriminating the effect of differences in the values of proinflammatory molecules according to the degree of OA progression

·       Line 98: please provide a definition for the abbreviation ACD-A.

Results

·       Cytokine analysis: I think that it would be easier for readers to understand if the same nomenclature is always used. For example, in line 134 you use the word “Recheck”, but in lines 138, 144 as well as in the figure you use “the day of suture removal”. The same happens with “day 0” (line 134) while in the figure you use the word “evaluation”. In my opinion this could confuse the readers, so I suggest you review all these terminology in the manuscript and try to always use the same ones.

·       Lines 135-137: In these lines you specify that the TNF-α ratio in the joint fluid of LrPRP injected dogs were significantly greater than the ration of control dogs on surgery day, however, in figure 1 it seems that the ratio in LrPRP group is lower than the ratio of control group. Can you please check this?

Discussion

·       Lines 177 and 178: “How often intra-articular injection of LrPRP must be repeated or how long a single injection is effective, are questions still not resolved” In my opinion this issue should be further discussed.

·       Line 179: why did you choose a 10-14 days interval to perform the LrPRP injections? Are there any references supporting this interval? Moreover, why not all dogs were injected with the same interval?

·       Lines 181-183: “Also, there is some controversy if intra-articular injection of PRP should be performed at the time of surgery or not, as intra-articular bleeding will be present and the effect of PRP may be limited” please provide a reference for this statement.

·       Line 206: The abbreviation LrPRP should be used instead of “leukoreduced platelet rich plasma”
